# Towards a nomenclature of health services for implementing universal health coverage in low- and middle-income countries

Alain Ndayikunda [1,2] ✉, Ronald Buyl [1] & Frank Verbeke[2,3]

## Abstract

**Background** Achieving Universal Health Coverage (UHC) in low- and middle-income countries (LMICs) requires a robust digital infrastructure capable of monitoring healthcare services and associated costs. A major barrier is the absence of a standardized and comprehensive nomenclature for billable healthcare services. Assessment across five hospitals in Burundi confirmed this problem by showing significant inconsistencies in service naming and coding. This study presents the development of a Universal Nomenclature of Health Services (UNHS) for Burundi, a meta-classification for billable health services, designed to align international classification systems with local operational needs.

**Methods** The methodology comprised a need assessment, literature review, the selection of relevant international standards, national adaptation, integration of operational sub-codes, and validation through stakeholder engagement.

**Results** The developed meta-classification based on six international standards (ICD-10-PCS, CPT, HCPCS, LOINC, RxNorm, and UB04) produces 82,433 codes covering 97.7% of health services relevant to UHC tracking.

**Conclusions** This paper details the methodology, structure, coverage, and implementation of the UNHS, offering a scalable model for improving health information system interoperability and UHC monitoring in LMICs.

## Plain language summary

Universal Health Coverage (UHC) is intended to ensure that all individuals have access to quality healthcare services without encountering financial barriers. In pursuit of this objective, numerous low- and middle-income countries (LMICs) have undertaken the digitization of their healthcare systems to enhance and monitor the delivery of health-care services. Nevertheless, the effective monitoring of service delivery in LMICs remains a complex undertaking. This complexity stems from healthcare software's inefficient data collection, often made worse by the use of incompatible, non-standard coding systems. The present study seeks to address these challenges by developing a comprehensive nomenclature for billable healthcare services. It provides a detailed account of the creation of a meta-classification, which is based on six widely recognized international classification systems. It also describes the implementation of the resulting nomenclature within the context of Burundi, where it has been validated and endorsed by the Ministry of Health. The adoption of the nomenclature is anticipated to enhance billing processes within healthcare facilities, promote interoperability among disparate systems, and facilitate effective monitoring of UHC initiatives.

[1]Department of Public Health, Faculty of Medicine and Pharmacy, VUB, Brussels, Belgium. [2]Digital Health Research Unit, Institut National de Santé Publique, Bujumbura, Burundi. [3]Department of Electronics and Informatics, Faculty of Engineering, VUB, Brussels, Belgium. ✉e-mail: alain.ndayikunda@vub.be

In the pursuit of health-related Sustainable Development Goals, numerous countries have initiated strategies to implement Universal Health Coverage (UHC), ensuring that all individuals can access quality health services without incurring financial hardship[1]. Despite this global commitment, achieving UHC remains challenging due to persistent fragmentation in service delivery and insufficient integration of health data systems for performance monitoring and policymaking[2].

Many LMICs have made progress in the digital transformation of their health systems, emphasizing the deployment of national data repositories, electronic supply chain systems, and digital health records[3]. Digitalization is widely acknowledged as a crucial enabler of UHC, offering improved access to health care and cost efficiency[4–8].

However, these digital advancements have introduced new complexities, particularly concerning system interoperability and the lack of a standardized nomenclature for healthcare services[9–12]. Interoperability issues stem from disparate health information systems that are unable to communicate effectively[13], while inconsistent coding standards hinder reliable data exchange. Despite efforts from the World Health Organization (WHO), a globally accepted ontology for healthcare services remains elusive[14].

Currently, healthcare facilities in LMICs predominantly rely on locally defined terminologies and coding frameworks, often developed in isolation and without adherence to international standards[15,16]. The resulting ecosystem is characterized by non-standardized, often incompatible service codes, which impedes data interoperability and undermines efforts to monitor UHC progress effectively[17,18]. Key challenges include reliance on unstructured data inputs, insufficient alignment with international billing classifications, and limited awareness of global standards among stakeholders.

To manage UHC effectively, it is critical to document each health service encounter in a structured and standardized format, capturing provider, patient, insurer, facility, and service data[19,20]. This requirement remains largely unmet across LMICs due to the abovementioned barriers.

Middleware platforms such as OpenHIE have shown promise in addressing interoperability challenges by standardizing data exchange frameworks[15,21,22]. While these platforms facilitate structured information sharing for health providers and facilities, they have not yet extended to standardizing billable service codes, a foundational element for UHC monitoring[7,8,20].

Burundi exemplifies both the digital progress achieved by low- and middle-income countries (LMICs) and the persistent gaps that remain. Approximately 95% of hospitals use digital health information systems, and efforts are underway to extend digitalization to primary healthcare facilities. Recently, the country adopted the OpenHIE framework to standardize data related to universal health coverage (UHC) at the national level. However, this transition has highlighted an urgent need for a national standardized nomenclature of health services, as outlined in the national eHealth plan[23].

Despite the introduction of OpenHIE, health information exchange has not yet become operational due to the absence of a shared and standardized language for healthcare service identification and billing across applications. As a result, only a single application is currently hosted on the OpenHIE platform for testing purposes. The lack of a comprehensive health services nomenclature adapted to the national context remains a critical barrier to interoperability and effective UHC implementation in Burundi.

This study focuses on the development of the Universal Nomenclature of Health Services (UNHS), a nomenclature rooted in international standards and tailored to the Burundian context, and explores its applicability to other LMICs for improving billing in healthcare facilities, supporting interoperability and strengthening effective monitoring of Universal Health Coverage.

## Methods
### Study setting
This study was conducted to support the implementation of Universal Health Coverage (UHC) in Burundi, a small, landlocked country in Central Africa. The project aimed to establish a standardized nomenclature of health services compatible with national billing practices and international standards.

### Need assessment for UNHS
Although the National Plan for the Development of Health Informatics had already highlighted the need for the country to have a standardized national nomenclature for healthcare services, a rapid assessment of the need for this nomenclature was conducted. It was carried out in three steps, summarized in the Fig. 1.

### UNHS development steps
The development of the Universal Nomenclature of Health Services (UNHS) followed a structured five-step process, illustrated in Fig. 2.

### Literature review
A comprehensive literature review was conducted to identify international classification systems capable of coding various components of the healthcare process[24–30]. These systems were analyzed for their applicability in forming an integrated nomenclature suitable for LMIC-contexts.

### Contextual adaptation to national needs
A selection of relevant codes was extracted based on Burundi's health system requirements. Missing codes were supplemented through national extensions to ensure comprehensive coverage of locally billable services.

Extensions are developed from the main code by appending a numerical suffix preceded by a dot. This structure clearly represents hierarchical dependence on the parent code in a logical and transparent manner. Each national extension code corresponds to a distinct billable health service and can be systematically positioned within the hierarchy under the corresponding principal code.

### Assessment of billing compatibility
The curated subset of codes was evaluated for its adequacy in meeting national billing and regulatory requirements.

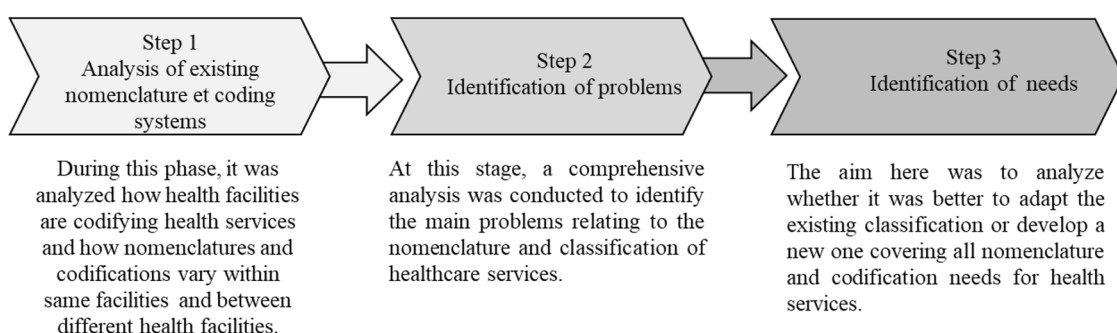

Fig. 1 | Steps of need assessment for UNHS.

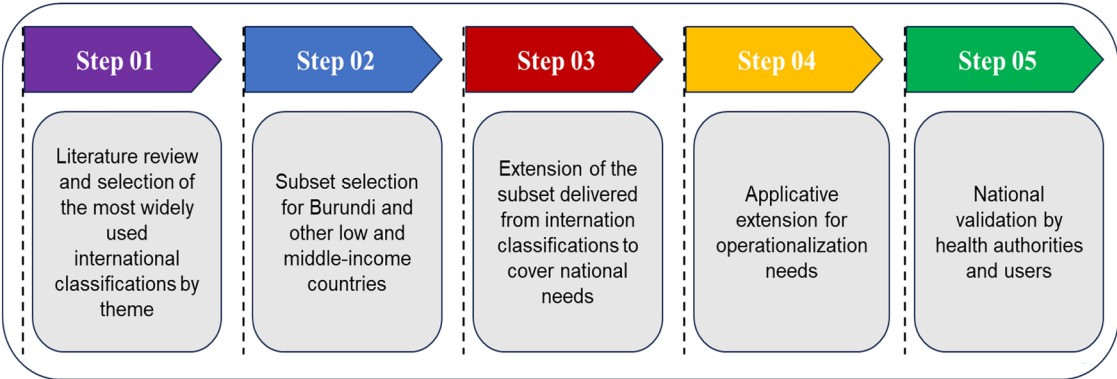

**Fig. 2 |** UNHS development and validation steps.

## Operational extensions

Sub-codes were developed to support practical applications within national health information systems such as Logistics Management Information Systems (LMIS). While not part of the core nomenclature, these sub-codes enhance operational usability.

## Validation by national stakeholders

The complete nomenclature underwent validation by key stakeholders, including government health authorities and medical societies representing various medical and surgical specialties. On invitation from the Ministry of Health, a total of 109 participants attended five UNHS validation workshops including 34 medical specialists, 43 nurses and technicians working in various fields, 3 pharmacists from the Burundian authority responsible for drug regulation, 2 pharmacists from the central drug purchasing agency in Burundi, 13 central-level health officials from the Ministry of Health, 6 technical officials working in drug logistics management in Burundi, 2 researchers from the health systems research unit at the National Institute of Public Health of Burundi and 7 experts working in the digitization of health facilities in Burundi. After the validation workshops, three meetings were organized for Ministry of Health executives to present the UNHS, and two UNHS conferences were held for health professionals. This validation step ensured alignment with professional practice and institutional expectations. It was carried out by researchers working with the program in charge of health system digitization at the Ministry of Health.

## Ethical approval

This study was approved by the Burundian National Ethics Committee. The approval number is CNE/12/2023.

## Results

### Needs assessment for UNHS

Results from the need assessment for UNHS, as described in methodology, revealed that only 33% of billable items had a common label across all facilities (Table 1). Additionally, every item (100%) had at least two different codes or labels used in two or more hospitals. Furthermore, 11% of the items had duplicate codes within the same hospital, while 9% of the items shared the same code despite representing different healthcare services, violating the principle of code uniqueness.

These results underscored the inadequacy of current service coding practices, confirming the need for a standardized, national nomenclature aligned with international frameworks.

### Literature review

A review of international literature identified twelve major domains of healthcare services relevant to both billing and UHC monitoring[24–30]. These domains include (i) Hospital procedures, (ii) Evaluation and management services (inpatient and outpatient), (iii) Medications, (iv) Prosthetics, (v) Consumables and medical devices, (vi) Transportation services, (vii)

### Table 1 | Assessment of local coding systems for health care billing

| Criterion | Local coding system |
| --- | --- |
| Consistent nomenclature across facilities | 33% |
| Duplicate codes in the same health facility | 11% |
| Multiple codes across different facilities | 100% |
| Same code for different services | 9% |

Hospital accommodation, (viii) Laboratory testing, (ix) Medical imaging, (x) Radiotherapy, (xi) Physical rehabilitation and diagnostic audiology and (xii) Mental health services. Each domain was selected for its importance for billing and its relevance to comprehensive UHC tracking.

### Selection of the most widely used international classifications in each domain

For each of the twelve identified domains, international classifications were reviewed. Six classification systems (Table 2) were selected based on the following criteria: (i) open access (when possible), (ii) broad international adoption and recognition, (iii) high granularity and specificity, (iv) regular updates to reflect evolving medical practices and (v) availability of detailed documentation[25–30].

Each classification was assigned a specific prefix (see Table 3) to maintain the link between the UNHS codes and their original international source.

### UNHS meta-classification coding algorithm

Each health service, with its associated international classification code, is being assigned a unique sequential numeric identifier. An alphanumeric UNHS code is then generated from this numeric identifier using a base-35 alphanumeric system (A–Z, 1–9). A final character is added as a check digit, computed by subtracting the modulo 26 of the numeric identifier of the UNHS code from 26, for internal consistency and verification.

Each character [A-Z] [1-9] has been assigned a value from 1 to 35 (Fig. 3). The numeric identifier that was assigned to "Drainage of Left Mastoid Sinus, Open Approach – ICD-PCS code 099COZZ" was **22718**.

The numeric value 22718 was converted to an UNHS code as follows:

- **Conversion to powers of 35**: $22718 = \mathbf{18} \times 35^2 + \mathbf{19} \times 35^1 + \mathbf{3} \times 35^0$
- **Conversion using the base 35 system**: $22718 = \mathbf{R} \times 35^2 + \mathbf{S} \times 35^1 + \mathbf{C} \times 35^0$
- **Checksum**: $26 - 22718 \bmod 26 = \mathbf{6}$, which corresponds to the letter **F**
- **Resulting UNHS code** = RSC + F = **RSCF**

The resulting UNHS contains 82,433 unique alphanumeric codes, each mapped to one of the six international classifications (Table 4). Labels for each code were provided in French, English, Spanish, and Portuguese, promoting multilingual usability.

**Table 2 | International classifications used to develop UNHS**

| Classification | Domain | Domain acronyms |
|---|---|---|
| ICD 10-PCS | Mental health, physical rehabilitation, hospital procedures, medical imaging, radiotherapy | MENT, PHY, PROC, IMG, RXT |
| CPT | Evaluation and management services | EVAL |
| HCPCS | Consumables, prosthetics, transportation | CONS, PROT, TRANS |
| LOINC | Laboratory | LAB |
| RxNorm | Medications | MED |
| UB04 | Hospital accommodation | LOG |

*ICD 10-PCS* International Classification of Diseases, 10th Revision, Procedure Coding System, *CPT* Current Procedural Terminology, *HCPCS* Healthcare Common Procedure Coding System, *LOINC* Logical Observation Identifiers Names and Codes.

**Table 3 | Prefix used for each international classification**

| Classification | Domain | Prefix |
|---|---|---|
| ICD 10-PCS | Mental health, physical rehabilitation, hospital procedures, medical imaging, radiotherapy | I |
| CPT | Evaluation and management services | C |
| HCPCS | Consumables, prosthetics, transportation | H |
| LOINC | Laboratory | L |
| RxNorm | Medications | R |
| UB04 | Hospital accommodation | U |

## UNHS subset selection for national needs

To tailor the nomenclature to Burundi's context, a comprehensive review was conducted of rate lists from 5 public hospitals, the national private practitioners' association, 3 health centers, 11 pharmacies, 2 physiotherapy centers and the National Reference Laboratory

This review produced 3298 distinct billable services for which coverage by international classifications was analyzed (Table 5). Of these:

- 91% were directly covered by international standards included in UNHS.
- 9% required national extensions (see example in Table 6), of which 6.7% were added as extensions mapped onto more generic header codes and 2.3% were operational-only codes without international equivalents (for local brand management, research or specific local care needs, e.g., "Care services related to the nose and sinuses").

To create an extension code, a numeric suffix is added to the main code, separated by a dot (Table 6).

To illustrate operational codes, as described in methodology, examples have been provided in Table 7.

## National UNHS subset validation

Following the inclusion of national extensions, the UNHS subset was evaluated for its effectiveness in covering local billing requirements. A control sample from five hospital databases, totaling 4,667 billing items, was used for validation. The UNHS successfully covered 4,465 of these items, representing a 95.7% coverage rate.

The "Physical Rehabilitation and Diagnostic Audiology" domain exhibited the lowest coverage, primarily due to heterogeneous billing practices across different health facilities. This variability suggests a need for harmonization through a nationally coordinated effort.

Further analysis revealed that the remaining unmatched items could often be mapped to more general existing UNHS codes. This allows facilities to retain detailed internal billing while still aligning with UNHS for standardized UHC monitoring.

## Sub-codes for specific application needs

To achieve near-complete coverage and improve operational usability, application-specific extensions were developed as sub-codes. These sub-codes respond to local implementation needs such as:

**Table 4 | Sample UNHS codes and mappings**

| UNHS Code | International code | Domain | Label |
|---|---|---|---|
| AYAXU | H-A0428 | TRAN | Ambulance service, basic life support, non-emergency transport |
| AX8DG | I-GZHZZZZ | MENT | Group psychotherapy |
| A6IGW | I-BW28ZZZ | IMG | CT scan of head |
| AINXD | I-0W9G3ZZ | PROC | Drainage of peritoneal cavity, percutaneous approach |
| AOSUV | H-A4207 | CONS | Sterile 2 cc syringe with needle |
| ASSZE | L-15074-8 | LAB | Glucose [Moles/volume] in Blood |
| ATSSI | R-315280 | MED | Aciclovir—200 mg tablet |

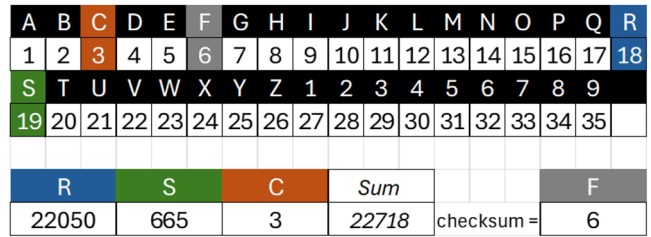

**Fig. 3 |** Example of UNHS code structure for "Drainage of Left Mastoid Sinus, Open Approach"

- Adapting billing to patient-specific financial categories
- Meeting health insurer requirements
- Capturing specific clinical details

For example, the ICD-10-PCS code "Plain Radiography of Lower Extremity" was found to be too general for Burundi, where imaging angles (e.g., sagittal, frontal) are frequently specified. Sub-codes were therefore created to capture this additional granularity.

## National validation by health authorities and users

The final validation process, as described in the methodology, was designed to facilitate adoption by all health system actors. Multidisciplinary review panels, including specialist physicians, nurses, pharmacists, and health IT developers, participated in an iterative review and refinement of the nomenclature's structure and content.

This collaborative validation led to minor adjustments in the meta-classification to better reflect real-world practices and ensure compatibility with the national system. The inclusive nature of the process increased stakeholder buy-in and laid the groundwork for national implementation.

**Table 5 | Coverage of local services codes by international classifications for a few domains**

| Domain | International coverage (%) | National extension needed (%) | Operational-only additions (%) | Overall coverage (%) |
|---|---|---|---|---|
| Drugs (RxNorm) | 93.6 | 6.4 | 6.4 | 93.6 |
| Laboratory (LOINC) | 98.9 | 1.1 | 1.7 | 98.3 |
| Hospital procedures (ICD 10-PCS) | 61.0 | 39.0 | 0.0 | 100.0 |
| Prosthetics (HCPCS) | 86.1 | 13.9 | 13.9 | 86.1 |
| Hospital procedures (ICD 10-PCS) | 99.1 | 0.9 | 0.8 | 99.2 |
| Overall | 91.0 | 9.0 | 2.3 | 97.7 |

**Table 6 | Examples of national extension codes**

| UNHS | International code | Domain | Label |
|---|---|---|---|
| AROEM | I-BW0CZZZ | IMG | Plain radiography of lower extremity |
| AROEM.1 | I-BW0CZZZ | IMG | Plain radiography of lower extremit—front view |
| AROEM.2 | I-BW0CZZZ | IMG | Plain radiography of lower extremity—front and profile views |
| AROEM.3 | I-BW0CZZZ | IMG | Plain radiography of lower extremity—profile view |
| AROEM.4 | I-BW0CZZZ | IMG | Plain radiography of lower extremity—other view |

**Table 7 | Example of operational codes without international equivalents**

| UNHS | Domain | Label |
|---|---|---|
| BCK9K | PROC | Services related to the larynx and trachea |
| BCLAJ | PROC | Services related to the neck |
| BCK7M | PROC | Services related to the nose and sinuses |
| BCK8L | PROC | Services related to the pharynx |
| BCKLH | PROC | Services related to the oral cavity |

## Discussion

This study demonstrates the feasibility of constructing a standardized nomenclature of billable health services for LMICs by leveraging existing international classification systems.

Today, several interoperability frameworks are available to promote health information exchange, structure, and standardization. However, current interoperability frameworks leave a nomenclature gap because they are often fragmented and have a non-harmonized code structure. UNHS fills this gap by providing a unifying framework on top of these fragmented classifications, bringing heterogeneous codes under a single reference system and thus offering a comprehensive common language for information exchange.

Using only six international standards, the initial UNHS meta-classification achieved 91% coverage of Burundi's health billing requirements. National extensions were incorporated to cover the remaining 9%, with only 2.3% of operationally relevant codes not matching any international standard. The remaining 6.7% were mapped to generalized international codes via header codes, preserving global interoperability while accommodating local billing needs.

These national extensions, although important for day-to-day operations, are of secondary relevance for high-level UHC monitoring. They can be abstracted or aggregated under broader UNHS categories, ensuring consistency across countries while retaining operational details at the local level.

Validation confirmed the applicability of the UNHS across Burundi's entire health system. As a result, the Ministry of Public Health has endorsed the UNHS for nationwide implementation. A pilot phase began on January 1, 2025, in five hospitals, with full-scale rollout scheduled for July 1, 2025. The nomenclature is openly available online[31], enabling other countries to consult and adapt the model.

Burundi's "green field" context, with minimal pre-existing national billing standards, provides an ideal environment for introducing a standardized nomenclature. In contrast, countries with entrenched legacy systems may face greater resistance or complexity when transitioning to a unified framework. Nonetheless, the absence of standardized billing classifications in many Sub-Saharan countries makes the UNHS highly scalable and potentially transformative for regional UHC monitoring and planning.

Several countries are implementing OpenHIE to promote interoperability between different digital solutions in their health systems. However, the experience in Burundi demonstrates that OpenHIE represents only a partial response to interoperability challenges, as it does not address the underlying problem of fragmented health service coding. The Universal Nomenclature of Health Services (UNHS) helps fill this gap by providing comprehensive, transversal, and standardized coding for billable health services, particularly in contexts where health facilities attempt to meet operational needs by creating non-standardized local codes.

UNHS has two major intrinsic strengths. First, its development benefited from a validation process involving a wide range of stakeholders, which guarantees not only the operational relevance of the nomenclature but also its suitability to LMIC field realities. Second, UNHS has national approval, which is a decisive strength. Both ensure its acceptability by users and its sustainability in application.

Despite its strengths, UNHS has several limitations. First, because it is derived from multiple international classifications, it requires continuous monitoring of changes across each source system. Maintaining synchronized monitoring is challenging, as updates to international classifications do not occur simultaneously, and the release of new versions may be delayed. Consequently, updates to UNHS inevitably lag slightly behind the most recent versions of its source nomenclatures. Second, existing international classifications do not adequately cover traditional medicine services, which are increasingly being integrated into health service delivery, particularly in Africa. To address this gap, UNHS has incorporated MAYELE codes, enabling more accurate representation of health services provided on the African continent. The introduction of UNHS carries significant political, practical, and research implications. Politically, it facilitates the monitoring of universal health coverage implementation by enabling the aggregation of health billing data from health facilities, something that was previously impossible. Practically, it provides the missing component for achieving interoperability among the various digital health billing solutions in use and is expected to enhance the overall quality of health system data. From a research perspective, UNHS creates new

opportunities to study health service provision. Collectively, these implications position UNHS as a potential lever for strengthening health system governance and improving efficiency.

## Conclusion

This study demonstrates that it is both feasible and beneficial to develop a comprehensive nomenclature of billable healthcare services for use in LMICs, based on a synthesis of internationally recognized classification systems. The resulting UNHS provides a standardized, interoperable framework that enables consistent documentation, billing, and monitoring of health services in support of UHC.

The Burundi case study illustrates how international standards can be adapted through national extensions to meet local regulatory and operational needs. While the model was tailored to Burundi, the methodology is transferable to other LMICs, especially those without existing national health service coding standards. Future research should assess the adaptability of the UNHS framework in additional countries and explore its integration into broader health information exchange platforms.

## Data availability

The Universal Nomenclature of Health Services (UNHS) dataset is available at https://unhs.mspls.org. DOI for UNHS is https://doi.org/10.5281/zenodo.18497226[31].

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

## Acknowledgements

The authors wish to thank the program in charge of health system digitization at the Ministry of Health in Burundi for its involvement during the validation process of UNHS.

## Author contributions

A.N. carried out the literature review, participated in the study design and drafted the manuscript. RB participated in the design of the study and closely reviewed the manuscript. F.V. carried out the literature review,

participated in the design of the study and closely reviewed the manuscript. All authors read and approved the final manuscript.

## Competing interests

The authors declare no competing interests
