## [Transparent Peer Review File · Communications Medicine]

Towards a nomenclature of health services for implementing universal health coverage in low- and middle-income countries

Corresponding Author: Dr NDAYIKUNDA Alain

Version 0:

Reviewer comments:

Reviewer #1

(Remarks to the Author)

The authors of 'Towards a nomenclature of health services for implementing universal health coverage in low- and middle-income countries' report on the implementation of digital healthcare system to the Burundi health service where the existing systems were ad hoc, inconsistent and unsuitable. The aim was to take existing international standards and adapt them to the specific needs of the Burundi and perhaps of further relevance to other LMICs.

This is a very clear description of the potential gains and the challenges needed to surmount in applying a nationally-relevant system that is adapted for the specific needs of the country without re-inventing the wheel.

There is clearly a significant amount of work that has gone into this transformational activity within the Burundi healthcare service which is of importance to other sub-saharan countries as well as LMICs in other parts of the world.

Main comments:

Overall the manuscript is well-written and flows well, however, the relationships between Tables 3, 4 & 5 are not clear. Specifically, in Table 4 is the code 'RSCF' the resulting code in UNHS or the source? How long are the UNHS codes? On line 134 I presume the reference to "one of the six international standards" refers to Table 3. If correct, please make explicit. The codes in Table 5 do not look like the example in Table 4. Should they? If not, why not?

On line 145-6 a reference is made to "research" operational-only codes without international equivalents. Such as what? No mention is made regarding how patients are tracked through the healthcare service. Do patients have a universal uniqueID or do they have multiple IDs throughout the service. Did this UNHS process identify any improvements/challenges with patient information.

A significant in countries with mature system is that they are poorly adapted for secondary uses such as population health research and/or service provision monitoring. Does the UNHS help address these issues? As mentioned by the authors their "green field" context is ideal to avoid this issues early. Likewise is the issue of harmonised laboratory data collection. Is there a plan to perform a thorough evaluation of the roll-out?

Minor comments:

- In Figure 1 the boxes under Step 3 and Step 4 need to hyphenate the words "classification" and "operationalization".
- Table 5 – NUPS needs to be spelled out.

Reviewer #2

(Remarks to the Author)

Brief summary of the manuscript

1. The manuscript builds a strong case to argue that creating and implementing universal nomenclature of health services (UNHS) could solved the common of fragmented service codes in LMIC which is a critical barrier to universal health care monitoring. By synthesizing six international classifications (ICD-10-PCS, CPT, HCPCS, LOINC, RxNorm and UB-04) the authors generate 82 433 codes, covering 97.7 % of services relevant to UHC tracking in Burundi. The nomenclature was

adopted by the national health authorities with a national pilot that started on 1 Jan 2025 and full roll-out planned for 1 Jul 2025

Overall impression of the work

2. It is a comprehensive work because it builds a nomenclature of health services from six international classifications, then the authors contextualize these nomenclatures to the Burundi case, then they validate it in multidisciplinary review panels with different stakeholders and, finally, they received an endorsement from the Ministry of Public Health.

Specific comments, with recommendations for addressing each comment

3. The case to develop a nomenclature could be stronger. In the introduction, the need to develop a national nomenclature comes from a paragraph (line 58) that does not have a reference. Even after reviewing references 9 to 12 that support the claim that there is a lack of standardized nomenclature, it is not clear that there is a need to develop a national nomenclature. This can be reinforced if, for example, in the introduction, the authors explicitly situate the nomenclature gap after the country implemented the OpenHIE platform and show that interoperability requirements are still unmet.

4. The first result in the manuscript, the Needs assessment for UNHS, was not mentioned in the methods section. There is no information on how this assessment was conducted.

5. The authors claim that they did a systematic literature review to inform the nomenclature. However, the paper that they cite is not a systematic review; it is a paper titled "Method for constructing a new extensible nomenclature for clinical coding practices in Sub-Saharan Africa" which was published in 2017 and it is not a systematic review. This paper would benefit from performing an evidence synthesis to identify how other national nomenclatures have been constructed.

6. There is also no information about how the authors conducted the validation process, how many people participated in the review panels? How were those panels conducted? What questions were asked?

7. A mayor revision of methods is strongly advised, the paper lacks enough detail about the methods, details needed include: a) inclusion/exclusion criteria for systematic review; b) validation panel composition; c) why domains were merged; and d) how national extensions were derived.

8. Regarding the discussion, most readers expect a clearer structure of the discussion section to be able to follow the argument (Results in context, strengths and limitations, implications). Some suggestions to strengthen this section include: 1) mapping findings against comparable studies, 2) state why existing interoperability frameworks still leave a "service coding gap" and how UNHS fills it, 3) the Introduction frames OpenHIE as only partially solving the , but the discussion does not circle back to show how UNHS completes that framework, 5) strengths are implied (wide stakeholder validation, national endorsement), but neither strengths nor limitations are explicitly enumerated. 6) Currently, the only implications statement is that UNHS is "potentially transformative" and "scalable", maybe adding policy, practice and research implications of this work will improve the article.

Reviewer #3

(Remarks to the Author)

I co-reviewed this manuscript with one of the reviewers who provided the listed reports. This is part of the Communications Medicine initiative to facilitate training in peer review and to provide appropriate recognition for Early Career Researchers who co-review manuscripts.

Version 1:

Reviewer comments:

Reviewer #1

(Remarks to the Author)

I thank the authors for the response to the review and I have no further comments to add.

Reviewer #2

(Remarks to the Author)

Thanks to the authors for addressing all of our comments. Our concerns were addressed. We share the importance of creating and implementing universal nomenclature of health services (UNHS) to solve the common of fragmented service codes in LMIC which is a critical barrier to universal health care monitoring.

Reviewer #3

(Remarks to the Author)

I co-reviewed this manuscript with one of the reviewers who provided the listed reports. This is part of the Communications Medicine initiative to facilitate training in peer review and to provide appropriate recognition for Early Career Researchers who co-review manuscripts.

Alain NDAYIKUNDA

Institut National de Santé Publique, Bujumbura, Burundi

Tel : +257 79 594 805

Email : alainndayikunda2014@gmail.com

December 27th, 2025

Subject: Response to reviewer's comments - revised manuscript (manuscript ID: COMMSMED-25-0814A)

Dear Editor-in-Chief,

We thank you and the reviewers for the time and effort devoted to the evaluation of our manuscript intitled "Towards a nomenclature of health services for implementing universal health coverage in low- and middle-income countries" (manuscript ID: COMMSMED-25-0814A). We are grateful for the constructive comments which have helped us to improve the quality of our manuscript.

We have carefully addressed all the comments raised by reviewers. Below, we provide a detailed, point-by-point response to each comment. Reviewers's comments are reproduced followed by our response. A version of our paper in track changes is provided to show you all changes to the initial version.

Reviewer #1

Comment 1: *Overall, the manuscript is well-written and flows well, however, the relationships between Tables 3, 4 & 5 are not clear. Specifically, in Table 4 is the code 'RSCF' the resulting code in UNHS or the source?*

Response: Commentary on Table 4 has been improved to ensure better understanding. (From line 186 to 192).

Comment 2: *How long are the UNHS codes?*

Response: Regarding length of UNHS code, we would like to explain that A UNHS code does not have a fixed length. Its length is determined by converting sequential numbers (1, 2, 3, ..., 1257, 1258, ...) into alphanumeric characters using the algorithm described in Table 4, to which a check digit is added to prevent coding errors during data entry. As a result, the length of the UNHS code is variable but cannot exceed 5 characters. In fact, by combining the 26 letters of the alphabet and the 9 digits (1-9), we find that we can generate more than 40 million different codes, which we will almost never reach in terms of different services.

Comment 3: *On line 134 I presume the reference to “one of the six international standards” refers to Table 3. If correct, please make explicit.*

Response: We thank the reviewer for highlighting this ambiguity. Indeed, we are referring to the six classifications in Table 3. We have added the reference in the text. The updated text is now on page number 7, just in the paragraph above Table 5.

Comment 4: *The codes in Table 5 do not look like the example in Table 4. Should they? If not, why not?*

Response: To clarify this point, we refer to the previous explanation on code length. The difference between these codes lies in the fact that the length of the codes may vary.

Comment 5: *On line 145-6 a reference is made to “research” operational-only codes without international equivalents. Such as what?*

Response: We thank the reviewer for pointing out this area needing clarification and improvement. We have revised the text and added table 7 where examples of operational codes without international equivalent have been provided. Table 7 and table 8 have been added to clarify this.

Comment 6: *No mention is made regarding how patients are tracked through the healthcare service. Do patients have a universal unique ID or do they have multiple IDs throughout the service. Did this UNHS process identify any improvements/challenges with patient information.*

Response: In response to this reviewer’s comment, we can say that UNHS was introduced at the same time with a pilot phase for the introduction of a unique patient identifier.

The improvements/difficulties have been implicitly provided in Table 1. UNHS primarily improves the tracking of services provided to patients by offering clear and distinct identification of services.

An assessment of the effects of UNHS implementation is planned for 2026. At that point, the improvements/difficulties will be clearly identified.

Comment 7: *A significant problem in countries with mature system is that they are poorly adapted for secondary uses such as population health research and/or service provision monitoring. Does the UNHS help address these issues?*

Response: The discussion has been updated to show how UNHS could help to address these issues. The last paragraph of the discussion has been added to answer to this question by showing some implications of UNHS.

Comment 8: *As mentioned by the authors their “green field” context is ideal to avoid these issues early. Likewise, is the issue of harmonized laboratory data collection. Is there a plan to perform a thorough evaluation of the roll-out?*

Response: Thank you again for your comment. UNHS includes a laboratory section based on LOINC codes. This will also harmonize the collection of laboratory data. A thorough evaluation of the rollout is planned after one year of use by healthcare facilities.

Comment 9: *Minor comments:*

- *In Figure 1 the boxes under Step 3 and Step 4 need to hyphenate the words “classification” and “operationalization”.*
- *Table 5 – NUPS needs to be spelled out.*

Response: We agree with your comment and have revised the manuscript accordingly. For NUPS, it was a mistake. It's UNHS. This has been corrected in the text. The modification can be checked in figure 1 and table 5.

Reviewer #2

Comment 1: *The case to develop a nomenclature could be stronger. In the introduction, the need to develop a national nomenclature comes from a paragraph (line 58) that does not have a reference. Even after reviewing references 9 to 12 that support the claim that there is a lack of standardized nomenclature, it is not clear that there is a need to develop a national nomenclature. This can be reinforced if, for example, in the introduction, the authors explicitly situate the nomenclature gap after the country implemented the OpenHIE platform and show that interoperability requirements are still unmet.*

Response: We appreciate this suggestion and have added further explanation in the introduction. Modifications can be found in paragraph 7 of the Introduction (line 74 to 79)

Comment 2: *The first result in the manuscript, the Needs assessment for UNHS, was not mentioned in the methods section. There is no information on how this assessment was conducted.*

Response: Thank you for this remark. Methodology section has been updated in response to this comment. A section describing the UNHS needs assessment process has been added to the methodology. See section 2 in methodology.

Comment 3: *The authors claim that they did a systematic literature review to inform the nomenclature. However, the paper that they cite is not a systematic review; it is a paper titled “Method for constructing a new extensible nomenclature for clinical coding practices in Sub-Saharan Africa” which was published in 2017 and it is not a systematic review. This paper*

would benefit from performing an evidence synthesis to identify how other national nomenclatures have been constructed.

Response: We have revised the manuscript to address this reviewer's concern. That was a mistake. We carried out a literature review but not a systematic review. This has been corrected in the text. The criteria for selecting the different nomenclatures were provided in the results section just before Table 2. The title of the first step of UNHS development stages has been changed from "Systematic literature review" to "literature review". See the text just after Figure 1.

Comment 4: *There is also no information about how the authors conducted the validation process, how many people participated in the review panels? How were those panels conducted? What questions were asked?*

Response: We appreciate this constructive comment. Additional informations have been included in methodology to address this point. This has been added to the methodology section, lines 128 to 141.

Comment 5: *A mayor revision of methods is strongly advised, the paper lacks enough detail about the methods, details needed include:*

- a) inclusion/exclusion criteria for systematic review;*
- b) validation panel composition;*
- c) why domains were merged; and*
- d) how national extensions were derived.*

Response: More informations have been provided in methodology to improve the reader's understanding.

We did not merge domains. As shown in Table 2, the domains presented are those covered by the classifications used. We did not merge these domains; we took them as they were in the classifications used because there are classifications that cover many domains.

Examples of national extension codes are given in Table 7, and the procedure for creating these codes is explained in the sentence preceding the table.

Updated text can be found at:

- Step 5 of UNHS development steps: validation by national stakeholders;
- Step 2 of UNHS development steps (Contextual adaptation to national needs)

Comment 6: *Regarding the discussion, most readers expect a clearer structure of the discussion section to be able to follow the argument (Results in context, strengths and limitations, implications). Some suggestions to strengthen this section include:*

- 1) mapping findings against comparable studies,*
- 2) state why existing interoperability frameworks still leave a “service coding gap” and how UNHS fills it,*
- 3) the Introduction frames OpenHIE as only partially solving the problem, but the discussion does not circle back to show how UNHS completes that framework,*
- 5) strengths are implied (wide stakeholder validation, national endorsement), but neither strengths nor limitations are explicitly enumerated.*
- 6) Currently, the only implications statement is that UNHS is “potentially transformative” and “scalable”, maybe adding policy, practice and research implications of this work will improve the article.*

Response: We appreciate the reviewer’s suggestions. All the suggestions have been included except for the first one as we didn’t find similar studies (on meta-classification of health services). This has been included in discussion, paragraph 2, 3, 8, 9, 10, 11.

We sincerely appreciate time and effort invested by the reviewers and the editorial team, and we hope that the revised manuscript will now be suitable for publication in “Communications Medicine”.

Thank you for your continued consideration,

Yours sincerely,

Alain NDAYIKUNDA